# The Effects of Viscoelasticity on Droplet Migration on Surfaces with Wettability Gradients

**DOI:** 10.3390/mi13050729

**Published:** 2022-04-30

**Authors:** Ying Jun Ren, Sang Woo Joo

**Affiliations:** School of Mechanical Engineering, Yeungnam University, Gyeongsan 38541, Korea; renyingjun2003@gmail.com

**Keywords:** droplet migration, viscoelasticity, wettability gradient

## Abstract

A finite-volume method based on the OpenFOAM is used to numerically study the factors affecting the migration of viscoelastic droplets on rigid surfaces with wettability gradients. Parameters investigated include droplet size, relaxation time, solvent viscosity, and polymer viscosity of the liquid comprising droplets. The wettability gradient is imposed numerically by assuming a linear change in the contact angle along the substrate. As reported previously for Newtonian droplets, the wettability gradient induces spontaneous migration from hydrophobic to hydrophilic region on the substrate. The migration of viscoelastic droplets reveals the increase in the migration speed and distance with the increase in the Weissenberg number. The increase in droplet size also shows the increase in both the migration speed and distance. The increase in polymer viscosity exhibits the increase in migration speed but the decrease in migration distance.

## 1. Introduction

The motion of liquid droplets on solid surfaces is ubiquitous in nature and is associated with extremely broad applications in many different fields [1,2]. The manipulation of droplets by controlling the wettability of substrates in particular has been intensively investigated due to daily observations and various potential industrial applications. The methods for wettability control for transporting droplets on surfaces include creating temperature gradients, electrowetting, magnetic fields, and chemical or physical texture gradients [3,4,5,6,7,8,9,10,11,12,13,14,15]. When droplets are placed on a solid surface with a wettability gradient, they tend to move from regions of low wettability to high wettability due to the net driving force in the direction of increasing surface wettability. Greenspan and Brochard [16,17] studied the wettability gradient of a surface to night drop operation in detail. The net driving force is due to the difference in curvature between the front and back half of the droplet. Yang et al. [18] further discussed on the concept of manipulating droplets in the absence of an external factor environment. Li and Zhiguang [19] verified the spontaneous motion of droplets on solid surfaces. Moumen et al. [20] performed detailed experiments on droplet transport on horizontal solid surfaces using wettability gradients. Chowdhury et al. [21] verified the droplet transport mechanism for wettability gradient trajectories with different constraints. Liu et al. [22] conducted experiments on the motion of droplets on a surface with a wettability gradient and found that the velocity of the droplets increased with the surface wettability gradient. Subramanian et al. [23] verified the forces involved in the migration of droplets on solid surfaces, and demonstrated the forces and resistance provided by droplets by approximating the shape of the droplets as wedges, which is known as the wedge approximation. They also used the lubrication approximation to study the dynamics and resistance of fluids. Chaudhry et al. [24] verified that the velocity gradually decreased after increasing to a certain value along the direction of droplet movement. Xu and Qian [25] analyzed the motion of nanoscale droplets on a heated solid surface with a wettability gradient, and accurately simulated the phenomenon of rapid changes in solid-to-fluid temperature. They investigated the motion of evaporative droplets in a single-component fluid on a solid substrate with a wettability gradient. There are two main difficulties with fluid flow and heat flow near droplet contact lines on solid substrates: hydrodynamic (stress) singularities and thermal singularities. A continuum hydrodynamic model is proposed for the study of the motion of single-component droplets of fluids on solid substrates. The model can handle thermal singularities, which inevitably arise as the substrate temperature differs from the coexistence of liquid and gas. Liu and Xu [26] performed theoretical analysis and molecular dynamics simulations of droplet transport on surfaces with wettability gradients. A unified mechanical model is proposed that integrates the static configuration of the droplet at equilibrium and the dynamic configuration of the droplet during motion. Molecular dynamics (MD) simulations show that the configuration of water droplets on a solid surface relaxes during motion, and a dimensionless parameter is proposed to describe their dynamic contact area. In addition, the analysis showed that the friction coefficient of water droplets on the solid surface was significantly different from that of the water film, and a geometric factor related to the wettability of the solid surface was formulated to calibrate the kinetic friction of water droplets. Full-velocity trajectories of droplet motion are extracted, and the predictions are in good agreement with extensive MD simulations across the entire surface wettability gradient, from superhydrophobic to superhydrophilic. Raman et al. [27] used a phase-field-based Boltzmann method (LMB) to simulate the dynamics of droplet aggregation on a wettability-gradient surface and observed that when the droplet impinges on the wettability gradient surface, the droplet shape is not necessarily spherical, resulting in different droplet morphologies near the droplet junction area. Huang et al. [28] conducted a 2D numerical simulation of droplet transport on a surface with a stepwise wettability gradient by considering the contact-angle hysteresis (CAH) on the droplet surface. They used the Lattice Boltzmann Method (LBM) and found that the velocity of the droplet has a strong dependency on viscosity ratio, wetting gradient magnitude, and CAH. Ahmadlouydarab and Feng [29] used wettability gradient and external flow to numerically study the movement and coalescence of droplets, and analyzed the transport of droplets on a surface with wettability gradient, making comparisons with the results of Moumen et al. [20] In addition to the work of Nasr et al. [30], Chaudhry et al. [21,31] studied the migration of droplets on surfaces with linear wettability gradients, and concluded that the droplet shape was found to evolve over time to maintain a minimum energy state. Even with different wettability gradients, the surface energy of a droplet can be the same at a specific dimensionless time. Droplets, located at different locations and times, can be identical in shape. Paul Ch. Zielke et al. [32] reported that velocity increases with the droplet size. 

Although the research on Newtonian fluids has made great progress, the research on non-Newtonian fluids (viscoelastic fluids) is still very scarce. In the study of non-Newtonian fluids [33,34,35,36,37,38] most of the studies are on flows in microchannels. Although the spontaneous migration of Newtonian droplets due to wettability gradients has been widely studied, that for viscoelastic droplets is very limited. Bai et al. [39] used the OpenFOAM to numerically analyze viscoelastic droplet migration on surfaces with linear wettability gradients. They showed that the migration speed increases monotonically with the increase in fluid elasticity until it saturates for high enough Weissenberg number. The migration distance, however, was not obtained because the cases reported did not contain migrations coming to an end, which is observed in experiments. Li et al. [40] proposed a dynamically controlled particle separation by employing viscoelastic fluids in deterministic lateral displacement (DLD) arrays. The process of deceleration and termination of viscoelastic droplet migration due to the growing viscous force with droplet deformation, however, is omitted in the report. Zhang et al. [41] studied the transient flow response of viscoelastic fluids to different external forces. Damped harmonic oscillation and periodic oscillation are induced and modulated depending on the fluid intrinsic properties such as viscosity and elasticity. External body forces, such as constant force, step force, and square wave force, are applied at the inlet of the channel. It is revealed that the oscillation damping originates from the fluid viscosity, while the oscillation frequency is dependent on the fluid elasticity. An innovative way is also developed to characterize the time relaxation of the viscoelastic fluid by modulating the frequency of the square wave force. Zhang et al. [42] investigated temporal-pulse flow mixing of Newtonian and viscoelastic fluids at different pulse frequencies and showed that viscoelastic fluids are more mixed than Newtonian fluids. Despite these findings on effects of the viscoelasticity, the finite migration distance of droplets affected by the viscoelasticity is yet to be reported. In this work the effect of viscoelasticity on the droplet dynamics is studied parametrically, and the changes in migration speed and distance due to viscoelasticity are revealed for the first time.

## 2. Numerical Simulation 

The volume-of-fluid (VOF) method is a simulation technique used to track and locate free-form surfaces or fluid interfaces in computational fluid dynamics. It uses static and migrating mesh to accommodate the evolution of the interface shape. It is based on the Eulerian formulation. In this paper, the VOF solver tracking interface included in OpenFOAM is used to calculate the volume fraction in the gas/liquid two-phase flow: α is the volume fraction of a liquid, and the value of α in the grid varies between 0 and 1. When a grid is completely filled with liquid, the value of α is 1. When there is no liquid in the grid, the value of α is 0. The continuity equation is then written as
(1)$$\frac{\partial \alpha }{\partial t}+\left(U\cdot \nabla \right)\alpha =0$$
where U is the fluid velocity vector. The transport properties of this fluid are obtained by a volume average of the equations:(2)$$\rho =\alpha \rho_{1}+\left(1-\alpha \right)\rho_{2}$$
(3)$$\mu =\alpha \mu_{1}+\left(1-\alpha \right)\mu_{2}$$
where ρ and μ represent the density and dynamic viscosity of the two liquids, respectively. The interactive reaction between the two phases of the fluid can be calculated on the surface tension by the following equation:(4)$$\Delta p=\sigma k\overset{⌢}{n}$$
where Δp is the pressure difference across the interface, σ represents the surface tension coefficient, k is the curvature of the surface, and $\overset{⌢}{n}$ is the unit outward normal on the surface. The surface tension is included in the Navier–Stokes equation as a source term. Based on the case of incompressible fluids, the governing equations of viscoelastic fluids and the conservation of mass and momentum can be expressed as
(5)Δ⋅U=0
(6)$$\frac{\partial \rho U}{\partial t}+U\cdot \left(\rho U\right)=-\nabla p+\nabla \left(\tau_{s}+\tau_{p}\right)+\sigma \kappa \nabla \alpha +\rho g$$
where p is the pressure, ρ is the fluid density, and g is gravitational acceleration, the stress tensor can be expressed as
(7)$$\tau =\tau_{s}+\tau_{p}$$
where stress τ is divided into that contributed by the Newtonian solvent $\tau_{s}$, and the viscoelastic polymer $\tau_{p}$. To focus on the effect of viscoelasticity without the complications of shear-thinning the Oldroyd-B viscoelastic constitutive model is adopted, which can be expressed as
(8)$$\tau_{p}+\lambda \tau_{p}^{\nabla }=2\eta_{p}\left[\nabla U+{\left(\nabla U\right)}^{T}\right]$$
where λ is the relaxation time, $\eta_{p}$ is the polymer viscosity, $\tau_{p}^{\nabla }$ is the derivative on the elastic stress tensor:(9)$$\tau_{p}^{\nabla }=\frac{\partial \tau_{p}}{\partial t}+\nabla \cdot \left(U\tau_{p}\right)-{\left(\nabla U\right)}^{T}\cdot \tau_{p}-\tau_{p}\cdot \left(\nabla U\right)$$

The relation of the Oldroyd-B constitutive model can be expressed as
(10)$$\tau_{p}=\eta_{p}/\left(\lambda \left(C-I\right)\right)$$
where C is the conformational tensor of the polymer molecule, and a symmetry tensor I is a unit tensor. Equation (6) can then be simplified as
(11)$$\frac{\partial U}{\partial t}+U\cdot \nabla U=-\frac{1}{\rho_{c}}\nabla p+\frac{\beta \eta_{c}}{\rho_{c}}\nabla^{2}U+\frac{\eta_{c}}{\rho_{c}\lambda_{c}}\left(1-\beta \right)\nabla \cdot C+\frac{\sigma k\nabla \alpha }{\rho_{c}}+g$$
where $\beta =\eta_{s}/\eta_{p}=\eta_{s}/\left(\eta_{s}+\eta_{p}\right)$ and $\eta_{c}=\eta_{s}+\eta_{p}$, The viscosity and density fields depend on the order parameter:(12)$$\eta_{c}=\alpha \eta_{L}+\left(1-\alpha \right)\eta_{G},\;\rho_{c}=\alpha \rho_{L}+(1-\alpha )\rho_{G}$$
where $\eta_{L}$ and $\rho_{L}$ denote the viscosity and density of the liquid. The viscosity and density of the gas is denoted by $\eta_{G}$ and $\rho_{G}$. The transport equation of the deformation rate tensor is expressed as
(13)$$\frac{\partial C}{\partial t}+\nabla \cdot \left(UC\right)-{\left(\nabla U\right)}^{T}\cdot C-C\cdot \left(\nabla U\right)=\frac{1}{\lambda }\left(1-C\right)$$

If we set the droplet radius as a and the total substrate length as L, and the dimensionless droplet radius in units of L is
(14)$$R=\frac{a}{L}$$

The spatial and temporal variables then are nondimensionalized as
(15)$$x^{*}=\frac{x}{L}\;,\;y^{*}=\frac{y}{L}\;,\;T=\frac{t\nu }{a^{2}}$$
where the kinematic viscosity $\nu =\eta_{c}/\rho_{L}$.

For viscoelastic droplets, the Weissenberg number Wi is an important parameter, the measure of fluid elasticity: (16)$$Wi=\frac{\nu \lambda }{a^{2}}$$

If we take the center point $x_{m}$ of the droplet as a reference for droplet location, dimensionless droplet location M and the dimensionless migration distance are expressed as
(17)$$M=\frac{x_{m}}{L}\;\mathrm{and}\;M_{f}=\frac{x_{f}}{L}$$
where $x_{f}$ is the final value of $x_{m}$ when the droplet ceases to move.

In this paper, the OpenFOAM software is used for computations. Initially a viscoelastic droplet is placed in a rectangular area with a length L of 10 mm and a height H of 1.5 mm. The contact angle along the substrate decreases from the superhydrophobic region in the left side to the hydrophilic region in the right side, and the droplet migration is observed as in Figure 1. The boundary conditions are set to no-slip on the substrate and atmospheric conditions on other boundaries, as available in the OpenFOAM. The spatial resolution of the calculation is determined by grid-independent studies to ensure an absolute error bound of 10^−6^ on the calculation of the droplet migration distance. The change in the contact angle decreases from 160° at the initial droplet location to 0° at the right end of the substrate. As shown in Figure 2, the deformation that occurs with droplet migration is consistent with [24].

The contact angle model we use is the dynamic contact angle model, and the equation of the dynamic contact angle model is:(18)$$\theta_{d}=\left\{ \begin{array}{l} \theta_{a},\;U_{w}\geq 0\\ \theta_{r},\;U_{w}\geq 0 \end{array}\right.$$
where $U_{w}$ is the velocity near the wall. The dynamic contact angle model based on OpenFOAM is
(19)$$\theta =\theta_{e}+\left(\theta_{a}-\theta_{r}\right)\tanh\left(\frac{U_{w}}{U_{\theta }}\right)$$
where $U_{\theta }$ is the characteristic velocity scale. $\theta_{a}$, $\theta_{r}$, $\theta_{e}$, respectively, are the advancing, the receding, and the balance angle. 

Density of the Oldroyd-B and the Newtonian liquid is set identically to ρ = 1000 kg/m^3^, while polymer and solvent viscosities are set to $\eta_{p}$ = 0.36 Pa∙s and $\eta_{s}$ = 0.04 Pa∙s, respectively. The initial relaxation time is set as λ = 0.01 s [43]. For the gas phase, we set ρ = 1 kg/m^3^ and $\eta_{s}$ = 1 × 10^−5^ Pa∙s, with the surface tension σ = 0.073 N/m, as listed in Table 1. The migration of droplets on the substrate with a contact angle distribution ranging from 160° to 0° is investigated.

Figure 2 shows representative cases of Newtonian and viscoelastic droplet migration obtained by the OpenFOAM simulations described above. The red and blue regions, respectively, represent liquid and air phase, with the arrows indicating the local velocity vector. As in Figure 1, the initial droplet shape is set to a semicircle, with which the droplet radius and volume can be clearly specified.

## 3. Results and Discussion

Figure 3a shows the time-dependent location of droplet center M for droplets with identical viscosity, as a droplet migrates from the superhydrophobic region to the hydrophilic region of the substrate. The slope of each line indicates instantaneous migration speed, which eventually becomes zero for all droplets shown. It is thus seen that droplets start to move due to the wettability gradient, decelerate, and cease to move due to the viscous dissipation. The migration speed in the early stages of the motion and the migration distance in the final stage both increase with the Wi. The time spent to reach the final stationary state is seen to decrease with Wi. It can thus be deduced that more elastic droplets migrate faster and farther and stop sooner. Figure 3b shows the location of droplets with different volumes, with an enlarged scale provided in the inset. With the fixed wettability gradient along the substrate, bigger droplets would experience bigger differences in the contact angle between the advancing and receding side of the droplet. It is thus seen that droplets with bigger initial radius migrate faster and farther. Since the difference in the migration distance is more pronounced than the migration speed, the time required to reach the stationary state increases as well with the initial droplet radius. 

Figure 4a shows the location of viscoelastic droplets with time, depending on the solvent viscosity $\eta_{s}$ for an identical polymer viscosity $\eta_{p}$. For high solvent viscosity, $\eta_{s}$ = 25 Pa∙s, the viscoelastic droplet migrates slowly and for a relatively short distance. Low solvent viscosity, $\eta_{s}$ = 0.04 Pa∙s, gives faster and longer migration, as can be easily understood. In Figure 4b three different polymer viscosities, $\eta_{p}$ = 0.36, 1.9, 4.2 Pa∙s, are tested with the solvent viscosity kept identically at $\eta_{s}$ = 0.04 Pa∙s. In early stages, the difference in migration speed is not conspicuous, but eventually the differences in migration distance and migration time are obvious. Droplets with higher polymer viscosity show shorter migration distance and smaller migration time. 

Figure 5a shows the dimensionless migration distance $M_{f}$ depending on different viscoelasticity. The final migration distance of the droplets increases with the Wi number. The Newtonian droplet, Wi = 0, shows shortest migration distance. The increase in the migration distance with the increase in Wi is monotonic. Near Wi = 80, the increase seems minimized, but the terminal equilibrated value for high Wi, if any, cannot be verified due to difficulties in computations for high enough Wi. As seen in Figure 3, a longer migration distance is accompanied by a shorter migration time. With the increase in Wi the total migration time decreases monotonically. The viscoelasticity seems to promote the droplet migration due to the wettability gradient both in terms of migration velocity and migration distance. The exact mechanism for this promotion needs to be analyzed. The increase in the migration distance with respect to the droplet radius is shown in Figure 5b. For all values of Wi, the migration distance increases monotonically with the droplet radius. In contrast to the migration-distance increase due to the elasticity, however, the total migration time also increases with the increase in the migration distance. It is to be noted that with in present length scale increase in the droplet radius is analogous to that in the wettability gradient. 

## 4. Concluding Remarks

Based on the VOF method, the effect of viscoelasticity on the migration distance, speed, and time for spontaneous droplet motion due to wettability gradients is analyzed for the first time. It is found that as the fluid elasticity increases, the farther and the faster the viscoelastic droplet migrates, but the sooner it stops. Increase in the droplet size also makes it migrate farther and faster, but total migration time becomes longer. Increase in the viscosity of the solvent causes the droplet to move more slowly and over a shorter distance, as does the change in the viscosity of the polymer due to elastic effects. The viscoelasticity seems to promote the droplet migration due to the wettability gradient both in terms of migration velocity and migration distance. The exact mechanism for this promotion needs to be analyzed. It is to be noted that in the present length scale used, increase in droplet radius is analogous to that in the wettability gradient. This work focuses on the effect of elasticity on the droplet migration without other complications of viscoelastic fluids, and so the Oldroyd-B model is chosen. With the features embedded in the OpenFOAM, it is straightforward to extend the work to other constitutive models with other desired effects. Here, the migration-promoting effect of elasticity is reported quantitatively. Its difference with the effect of wettability-gradient increase is revealed.

## Figures and Tables

**Figure 1 micromachines-13-00729-f001:**
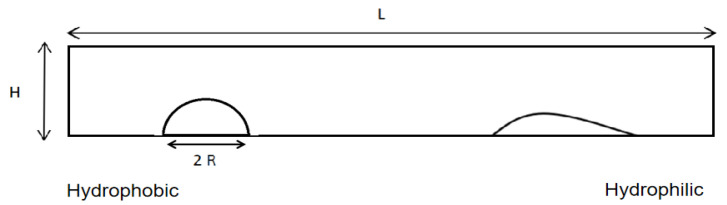
The computational domain of the simulation. Semicircle: The droplets move from the superhydrophobic side to the hydrophilic side.

**Figure 2 micromachines-13-00729-f002:**
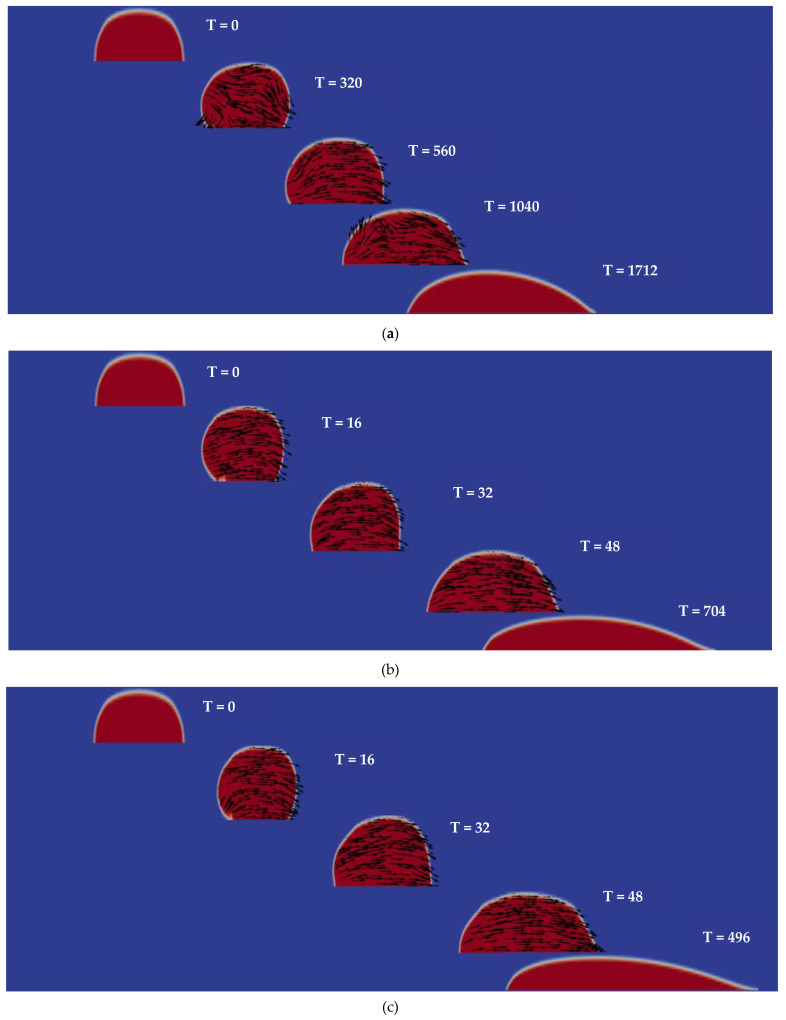
(**a**) Migration of a Newtonian droplet (**b**) Migration a of viscoelastic droplet with Wi = 16 (**c**) with Wi = 40.

**Figure 3 micromachines-13-00729-f003:**
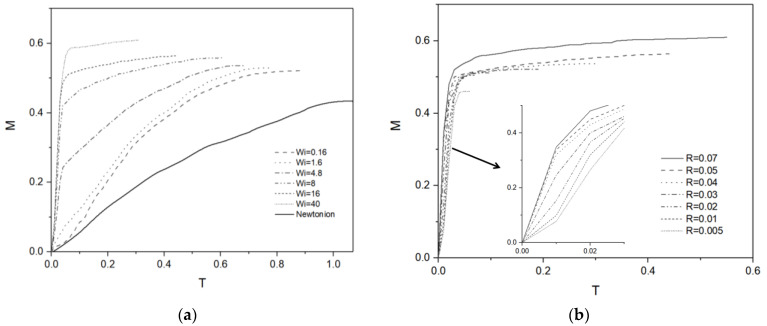
(**a**) Migration of Newtonian droplet and viscoelastic droplets with different Wi numbers against dimensionless time. (**b**) Migration of a viscoelastic droplet (Wi = 16) of different initial radii.

**Figure 4 micromachines-13-00729-f004:**
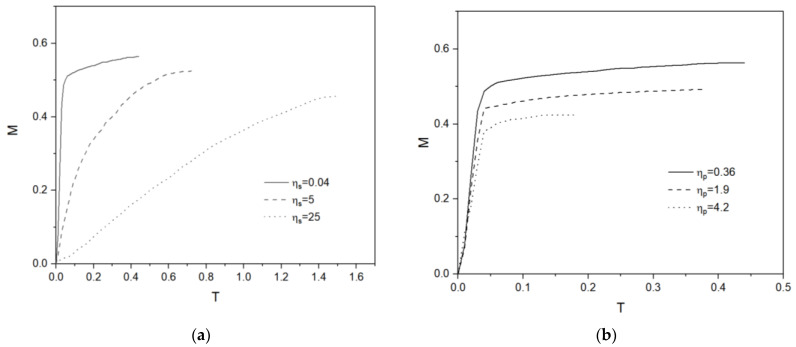
(**a**) Droplet migration for different $\eta_{s}$. (**b**) Droplet migration for different $\eta_{p}$.

**Figure 5 micromachines-13-00729-f005:**
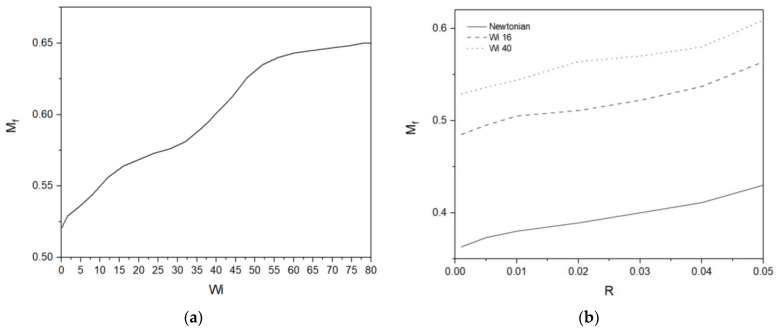
(**a**) Migration distance of droplets for different Wi; (**b**) Migration distance of droplets with different initial radius.

**Table 1 micromachines-13-00729-t001:** Liquid properties used.

Fluid	ρ (kg/m^3^)	$\eta_{p}$ (Pa∙s)	$\eta_{s}$ (Pa∙s)	λ (s)	σ (N/m)
Oldroyd-B	1000	0.36	0.04	0.01	0.073
Newtonian liquid	1000		0.04		0.073
Newtonian gas	1		$1\times 10^{-5}$		0.073

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
