# Peer review of "The Effects of Viscoelasticity on Droplet Migration on Surfaces with Wettability Gradients"

_micromachines, 2022, doi:10.3390/mi13050729_

Round 1

Reviewer 1 Report

In the manuscript entitled as “The Effects of Viscoelasticity on Droplet Migration on Surfaces with Wettability Gradients", the authors studied the migration of viscoelastic droplet on a rigid surface with wetting gradients using a  finite-volume method based on the OpenFOAM. The results seems interesting, while the authors didn’t present their work clearly. Besides, there are a lot of typo and I strongly suggest the authors to Major Revise the manuscript. Below are some detailed comments and questions:

  1. Line 81 claims the model is a two-phase flow, did the author mean there are two types of viscoelastic liquid, or just one liquid having interface with air?
  2. The author need to state the meaning of Vin equation 1.
  3. Line 101, “p is the fluid density”, should be rho.
  4. Line 112, the “expressed as.”should be “expressed as:”
  5. Could the author explain equation 14? why the length affect the radius of the droplet
  6. Line 120, “radius as a”, the “a”should be italic, or better change to another symbol
  7. In the model, I didn’t see the setting of the wettability gradients, is it due to the hydrophobic and hydrophilic change in a distance of L? if so, how to quantify the gradients? And the did the author tested the effect of different wettability gradients?
  8. I can’t understand the explain of equation 17, the author may use a figure to explain
  9. Line 189, “are tested anidentical solvent viscosity”, what does this mean?
  10. The authors presented their calculated results, but should discuss the reason on the migration change  at different parameters.

In conclusion, I suggest a major revise for this manuscript.

Author Response

Thank for your comments.

Reviewer 2 Report

This manuscript investigated the effects of droplet size, relaxation time, solvent viscosity, and polymer viscosity on the migration. I think this is a good work and I believe that the topic that covered in this manuscript is appropriate for publication in micromachines. I have the following major comments for the author to consider prior to publication. 

1. Author needs to more clarify the differences between the author’s previous work [32] and current work in the introduction. 
2, Comparison between obtained results and literature data is the very weakness of this work. For more contribution, the Authors should compare their results with those in relevant published works of other researchers. If it is possible, the authors should compare their results with those of previous works.
3. Author set the initial contact angle to 160 deg. However, Fig. 2 does not show the superhydrophobicity at T = 0.
4. In Table 1, the unit of surface tension is wrong. And why is the solvent viscosity of Newtonian (water) 0.04 Pa s in the table?
5. What does the arrow means in Fig. 3b?
6. This manuscript presents a lot of English gramma mistakes and there is not a good cohesion which makes complex to understand what the authors are clamming. Check the syntax error in the manuscript before its publication.

Author Response

Thank for your comments.

Reviewer 3 Report

The authors carried out an numerical investigation on the migration characteristics of viscoelastic droplet based on VOF method. The topic itself did attract my attention. However, the manuscript lacks in depth explanation with mostly the description of results (fig.3-5). The exploration seems rather straightforward. More systematic discussion should be provided. More importantly, the authors present the results without proper explanation, hence the novelty appears limited to me. The authors should try to focus on the mechanisms behind. The presenting and writing should also be improved. Therefore, I suggest rejecting the manuscript.

The listed are some suggestions for the authors to consider:

  1. Please include a proper comparison for verification purpose and also help to improve the presentation of the velocity field. The current form is unacceptable to me.
  2. Please help to include the mesh independent study in the SI.
  3. Can the authors help to explain the trend in fig. 5?
  4. Can the authors indicate the reference of table 1 line161-165, especially the relaxation time?
  5. Missing period p.g 1 line. 21, line 26
  6. What does the author mean by “further along ” p.g 1 line 43
  7. Line 46, can the authors explain a bit more on the term “solid-to-fluid temperature”?
  8. Line “Liu and Xu [28] performed theoretical analysis 47 and molecular dynamics simulations of droplet transport on surfaces with wettability gradients.” And what then?
  9. Please revise the phrasing line 55, “d found that the droplet velocity to viscosity ratio, wettability-gradient degree, and CAH are dominant parameters.”
  10. Line 64, it is a bit confusing to me when the authors write “Although the droplet is found at a different location at 64 a fixed time, the shape of the droplet is found identical.” Please help to revise.
  11. Please include the definition of a in eq.(14) and rewrite eq. (15)
  12. Please clarify “atmospheric boundary condition on other boundaries” line 138.
  13. Please double check line 155
  14. The authors should note that there should be a space between the number and the unit.
  15. Please redraw the inset of figure 3 (b) as there is no proper scale.

Author Response

Thank for your comments.

Round 2

Reviewer 1 Report

The revised version of the manuscript has well addressed the previous comments, but in fact the viscoelastic liquid has many other novel properties such as resonant response to the external force, or improving the mixing effect under micro-scale condition, for which following references can be added in the introduction part:

“Modulation of viscoelastic fluid response to external body force”,  Scientific Reports 9, 9402 (2019) 

“Comparison of Micro-Mixing in Time Pulsed Newtonian Fluid and Viscoelastic Fluid”,  Micromachines ,10, 262 (2019) 

In conclusion, I support the acceptance of the manuscript after a minor revise.

Author Response

Thank for your comments.

Reviewer 2 Report

All the issues were resolved in the revised version.

Author Response

Thank for your comments.

Reviewer 3 Report

The authors should conduct a more in-depth discussion on the topic which would be interesting to me. Please help to improve figure2 as there isn't much information upon the velocity field, which failed to show the flow field patterns with proper scales.  Please also help to relase more details for the mesh and the mesh independent study (eg. how the comparision is made and in which perspective you made the comparision). The authors should use different symbols for the variables before and after the non-dimensionlization (line 151).  

Author Response

Thank for your comments.
